# Avocado Oil Extract Modulates Auditory Hair Cell Function through the Regulation of Amino Acid Biosynthesis Genes

**DOI:** 10.3390/nu11010113

**Published:** 2019-01-08

**Authors:** Youn Hee Nam, Isabel Rodriguez, Seo Yeon Jeong, Thu Nguyen Minh Pham, Wanlapa Nuankaew, Yun Hee Kim, Rodrigo Castañeda, Seo Yule Jeong, Min Seon Park, Kye Wan Lee, Jung Suk Lee, Do Hoon Kim, Yu Hwa Park, Seung Hyun Kim, In Seok Moon, Se-Young Choung, Bin Na Hong, Kwang Won Jeong, Tong Ho Kang

**Affiliations:** 1Department of Oriental Medicine Biotechnology, College of Life Sciences and Graduate School of Biotechnology, Kyung Hee University, Global Campus, Gyeonggi 17104, Korea; 01030084217@hanmail.net (Y.H.N.); isabelula3r@gmail.com (I.R.); wanlapa.nuankaew@gmail.com (W.N.); caty05@naver.com (Y.H.K.); rodrigocastanedaqf@gmail.com (R.C.); tjdbf26@gmail.com (S.Y.J.); 01026793977@hanmail.net (M.S.P.); habina22@hanmail.net (B.N.H.); 2Gachon Institute of Pharmaceutical Sciences, College of Pharmacy, Gachon University, Incheon 21936, Korea; syjeong46@naver.com (S.Y.J.); phamnguyenminhthud12@gmail.com (T.N.M.P.); 3R&D Center, Dongkook Pharm. Co., Ltd., Gyeonggi 16229, Korea; lkw1@dkpharm.co.kr (K.W.L.); ljs@dkpharm.co.kr (J.S.L.); Kdh2@dkpharm.co.kr (D.H.K.); pyh@dkpharm.co.kr (Y.H.P.); 4College of Pharmacy, Yonsei Institute of Pharmaceutical Sciences, Yonsei University, Incheon 21983, Korea; kimsh11@yonsei.ac.kr; 5Department of Otorhinolaryngology, Yonsei University College of Medicine, Seoul 03722, Korea; ISMOONMD@yuhs.ac; 6Department of Preventive Pharmacy and Toxicology, College of Pharmacy, Kyung Hee University, Seoul 02453, Korea; sychoung@khu.ac.kr

**Keywords:** avocado oil, sensorineural hearing loss, hair cell, zebrafish, mice, HEI-OC1

## Abstract

Sensorineural hearing loss (SNHL) is one of the most common causes of disability, affecting over 466 million people worldwide. However, prevention or therapy of SNHL has not been widely studied. Avocado oil has shown many health benefits but it has not yet been studied in regards to SNHL. Therefore, we aimed to investigate the efficacy of avocado oil on SNHL in vitro and in vivo and elucidate its mode of action. For the present study, we used enhanced functional avocado oil extract (DKB122). DKB122 led to recovery of otic hair cells in zebrafish after neomycin-induced otic cell damage. Also, DKB122 improved auditory sensory transmission function in a mouse model of noise induced-hearing loss and protected sensory hair cells in the cochlea. In addition, RNA sequencing was performed to elucidate the mechanism involved. KEGG pathway enrichment analysis of differentially expressed genes showed that DKB122 protected House Ear Institute-Organ of Corti 1 (HEI-OC1) cells against neomycin-related alterations in gene expression due to oxidative stress, cytokine production and protein synthesis.

## 1. Introduction

Sensorineural hearing loss (SNHL) is a major disease worldwide that is categorized as age-related, noise-induced or idiopathic. SNHL accounts for approximately 90% of reported hearing loss [1,2,3,4,5]. Over 466 million people are living with disabling hearing loss worldwide, and this number is projected to increase to 933 million by 2050 (World Health Organization, 2018). However, prevention or treatment of hearing loss has not been widely studied. Thus, we have been searching for foods with potential to treat SNHL. After screening several candidates, we identified avocado oil as a promising functional food.

Avocado (*Persea americana* Mill.) is a popular tropical fruit that is cultivated in tropical and Mediterranean climates. The fruits are high in fatty acids, fiber, potassium, vitamin B3, and bioactive compounds such as vitamin E, carotenoids, and sterols [6,7]. Avocado has broad functional benefits, including anti-cancer, anti-inflammatory, anti-oxidant, and anti-microbial activity [8]. Specifically, avocado oil is rich in monounsaturated fatty acids, which are good for human health, and is considered a functional food [7,9,10]. Many beneficial effects of avocado oil have been reported; however, little is known about the potential efficacy of avocado oil on SNHL. Therefore, we aimed to investigate the efficacy of avocado oil on SNHL in vitro and in vivo and to elucidate its mode of action.

For the present study, we used enhanced functional avocado oil extract (DKB122). Further, we evaluated the recovery of otic hair cell function by DKB122 on neomycin-induced otic hair cell damage in zebrafish and investigated the viability of a cochlear organ in a Corti-derived cell line (HEI-OC1). In addition, we evaluated the effects of DKB122 in a mouse model of noise induced-hearing loss (NIHL) by assessing the auditory brainstem response (ABR) and viewed cochlear hair cells through rhodamine phalloidin staining. To further investigate this mechanism, we then performed RNA sequencing and evaluated the Kyoto Encyclopedia of Genes and Genomes (KEGG) pathway via an enrichment analysis of differentially expressed genes (DEGs). Then, we analyzed the effects of DKB122 treatment on the expression of up-regulated genes *Foxo1*, *Tgfb2*, and *Gadd45g* and down-regulated genes *Cth*, *Psph*, *Psat*, and *Mthfd2* by neomycin. In this study, we demonstrated that DKB122 is efficacious in treating hearing loss by protecting against altered gene expression due to oxidative stress, cytokine production and protein synthesis.

## 2. Materials and Methods

### 2.1. Sample Preparation

DKB122 is an avocado oil extract, and was provided by Dongkook Pharm. Co., Ltd., (Suwon, Korea). The Lot number is BDP180313-3. More than 56% of DKB122 is composed of four main fatty acids (palmitoleic acid, linoleic acid, palmitic acid and oleic acid). DKB122, which is from crude avocado oil, was manufactured through ion exchange resin with isopropyl alcohol. Briefly, the preparation of the column was performed by soaking resin in 50% ethanol, and then packing it into a 3.5 cm diameter column up to 25 cm high. Avocado crude oil was dissolved in 50% ethanol, then loaded into the column, until the oil was absorbed into resin. Then, it was successively partitioned with water, 50% ethanol, and isopropyl alcohol. Each of the residues were weighed after removal of the solvents in vacuo.

### 2.2. Animals

Wild-type adult zebrafish (*Danio rerio*) were housed in an S type 1500(W) × 400(D) × 2050(H) mm zebrafish system (WoojungBio, Inc., Suwon, Korea). Three pairs of zebrafish were put overnight in a spawning box and eggs were collected 3 h post-fertilization. Eggs were then incubated in petri dishes with 0.03% of sea salt solution prepared from sea salts purchased from Sigma-Aldrich Co. (St. Louis, MO, USA). Embryos were maintained under a 14 h light/10 h dark cycle in an incubator at 28.5 °C ± 5.0 until 6 days post-fertilization (dpf) when the experiments were performed.

Six-week-old Institute of Cancer Research (ICR) male mice were purchased from Orient Bio, Inc. (Seongnam, Korea). Mice were housed under a 12 h light/dark cycle with food and water provided ad libitum and maintained at controlled temperature (23.0 ± 2.0 °C) and humidity (50.0 ± 5.0%). Following a week of acclimation, mice were evaluated using the auditory brainstem response (ABR) before noise exposure to confirm normal hearing (≤25 dB).

### 2.3. Ethical Statement

All zebrafish experimental procedures were carried out in accordance with standard zebrafish protocols and were approved by the Animal Care and Use Committee of Kyung Hee University [KHUASP(SE)-15-10]. All experimental procedures using mice were carried out in accordance with protocols approved by the Animal Care and Use Committee of Kyung Hee University [KHUASP-15-17].

### 2.4. Neomycin-Induced Ototoxicity in a Zebrafish Model

To induce ototoxicity in zebrafish, wild-type zebrafish larvae were placed into a 96-well plate and treated with 100 µL of 2 µM neomycin sulfate (MB Cell Co., Seoul, Korea) for an hour.

To evaluate the effect of crude avocado oil (AVM) or DKB122 on otic hair cell recovery, the neomycin solution was removed and zebrafish were rinsed with 0.03% sea salt solution. Then the zebrafish were exposed to AVM 10 μg/mL or DKB122 10 μg/mL for 8 h at 28 °C. After the treatment, the zebrafish were rinsed with 0.03% sea salt solution and stained with 0.1% YO-PRO-1, purchased from Thermo Fisher Scientific Inc. (Ganseville, FL, USA) for 30 min, then anesthetized with 0.04% tricaine. The otic neuromast (O1) hair cells were then counted after visualization under a fluorescence microscope (Olympus 1 × 70; Olympus Co., Tokyo, Japan). All images were analyzed by Focus Lite software (Focus Co., Daejeon, Korea).

### 2.5. The 50% Effective Concentration (EC_50_)

Zebrafish were treated with six different concentrations (0.01, 0.1, 1, 10, 25 and 50 μg/mL) of DKB122. The EC_50_ values were calculated by non-linear regression using GraphPad Prism version 5.01 software (Graph Pad Software, San Diego, CA, USA).

### 2.6. Noise-Induced Hearing Loss (NIHL) in Mice

The experimental mice were exposed to 115 dB sound pressure level (SPL) as a broad band noise with a frequency spectrum from 100 Hz to 10 kHz for 1 h 30 min to induce hearing loss. During noise exposure, the mice were kept in a cage inside a noise chamber (250 cm × 560 cm × 550 cm) at 20–23 °C. The total noise level was measured at the center of the cage using a sound level meter SL-5868P (Hanada Technology Co. LTD, Zhejiang, China).

The mice were divided into 3 groups (*n* = 10/group) and treated orally once daily as follows: control mice with 0.3 mL of vehicle (80% distilled water, 10% dimethyl sulfoxide (DMSO), and 10% Tween 20), 100 mg/kg DKB122 in the vehicle (DKB122, 100 mg/kg), and 300 mg/kg DKB122 in the vehicle (DKB122, 300 mg/kg). The treatment started 1 day after noise exposure and continued up to 20 days.

### 2.7. Auditory Brainstem Response (ABR) Test

Auditory function tests were performed using channel recording (Intelligent Hearing Systems, Miami, FL, USA). Animals were anesthetized using xylazin (Bayer, Leverkusen, Germany), ketamine (Yuhan Corporation, Seoul, Korea), and saline solution (JW Pharmaceuticals, Seoul, Korea) (1.1:4:4.9, respectively) administered intramuscularly before the auditory function test. Mice were placed in an electrically and acoustically shielded sound attenuation booth TCA-500D (Sontek, Paju, Korea). The stimuli were delivered via earphones (Etymotic ER-EA). Needle electrodes were placed subcutaneously at the vertex of the skull, in the postauricular region and in the lower back. The auditory electrode needle resistance was 2–5 kΩ. Body temperature was maintained at 37 ± 1 °C using a heat lamp during the auditory test in order to protect from hypothermia.

Hearing thresholds were measured using an auditory brainstem response (ABR) at 1, 10, and 20 days after noise exposure. For the ABR recordings, clicks, 8 kHz and 16 kHz stimuli were performed for the analysis of ABR. Alternating clicks stimuli (0.1-ms duration) were delivered through earphones (Etymotic ER-3A) at a rate of 20.1 stimuli/s. For 8 kHz and 16 kHz tone bursts (rise-plateau-fall; 2-1- 2 cycles) were delivered through high frequency transducers. Physiological filters were set to pass electrical activity between 100 and 3000 Hz. Monaural responses were recorded for each mouse and averaged across a 10.24-ms time window. One thousand sweeps were collected. To determine the thresholds of ABR recordings, either the clicks, 8 kHz or 16 kHz were reduced in 10 dB steps, and then 5 dB steps as we approached the threshold.

### 2.8. Evaluation of Otoprotective Effects of Cochlear Hair Cells

Twenty days after drug administration, mice cochleae were harvested and perfused with 4% paraformaldehyde in phosphate-buffered saline (PBS) for 12 h at 4 °C. After decalcification with 0.1 M EDTA for 14 days, the organs of Corti were microdissected under a microscope as a surface preparation. The hair cells were evaluated on the animals in four groups: control, NIHL, and NIHL + DKB122 (n = 6 per group). The cochlea was stained with 5 U/mL rhodamine phalloidin to identify cochlear hair cells. Hair cells were examined under the microscope. The residual outer hair cells (OHCs) were counted in a 1-mm strip.

### 2.9. Cell Culture

Cells from the House Ear Institute-Organ of Corti 1 (HEI-OC1) mouse auditory cell line were cultured under permissive conditions (33 °C with 10% CO_2_) with high-glucose Dulbecco’s Eagle’s medium (DMEM, Sigma-Aldrich Co., St. Louis, MO, USA) containing 10% fetal bovine serum (FBS; WELGENE Inc., Gyeongsangbuk-do, Korea) and 50 U/mL INF-γ (Peprotech Inc., Seoul, Korea) without antibiotics as previously described [11].

The cells were subcultured in a density of 1 × 10^4^ cells/well in 96-well flat bottom plates. The cells were incubated for 24 h and then pre-treated for 1 h with DKB122 in a final concentration of 0.1–5 μg/mL, followed by cotreatment of the same concentration of DKB122 with 10 mM neomycin for 24 h. After the treatment at 33 °C, cells were exposed to 0.5 mg/mL of MTT (Duchefa Biochemie, Amsterdam, The Netherlands) solution for 4 h. After incubation, the solution was removed and 100 μL of DMSO were added to each well to solubilize the formazan crystals. Absorbance was measured in control cells was taken as 100% viability.

### 2.10. mRNA Sequencing and Pathway Analysis

HEI-OC1 cells were plated at a density of 4 × 10^4^ cells/well in a 6-well plate. Cells were pretreated with DKB122 (5~40 μg/mL) for 1 h and then neomycin was added (10 mM). After 24 h, cells were washed using PBS. Total RNA was extracted using TRIzol RNA Isolation Reagent (Invitrogen, Carlsbad, CA, USA) and then further purified using an RNeasy mini kit (QIAGEN, Hilden, Germany) to remove genomic DNA.

The quantity and quality of total RNA were evaluated using an Agilent 2100 Bioanalyzer (Agilent, Santa Clara, CA, USA). The isolated total RNA was processed to prepare a sequencing library using the TruSeq Stranded mRNA Sample Preparation kit (Illumina, San Diego, CA, USA) according to the manufacturer’s protocol. The quality and size of the libraries were assessed using an Agilent 2100 bioanalyzer (Agilent). All libraries were quantified by qPCR using the CFX96 Real Time System (Bio-Rad, Hercules, CA, USA) and sequenced on a NextSeq500 sequencer (Agilent) with a paired-end 75-bp plus single 8-bp index read run. KEGG analysis was performed using EnrichR [12].

### 2.11. Real Time-qPCR

After isolation of total RNA as described above, cDNA was synthesized using an iScript cDNA synthesis kit (Bio-Rad). Real time qPCR was performed using a Roche LightCycler 480 (Roche, Indianapolis, IN, USA). The sequences of primers used for RT-qPCR were: *Cth*, 5′-tgctaaggccttcctcaaaa-3′ (forward) and 5′-gtccttctcaggcacagagg-3′ (reverse); *Psph*, 5′-agatggagctacggacatgg-3′ (forward) and 5′-gtaccacttggcgttgtcct-3′ (reverse); *Psat1*, 5′-cagtggagcgccagaatagaa-3′ (forward) and 5′-cctgtgccccttcaaggag-3′ (reverse); *Mthfd2*, 5′-acagatggagctcacgaacg-3′ (forward) and 5′-tgccagcggcagatattaca-3′ (reverse).

### 2.12. Statistical Analysis

Data were analyzed using GraphPad Prism (version 5) statistical software package (GraphPad, San Diego, CA, USA). All data are expressed as means ± standard error of the mean (SEM). The statistical significance of the differences between groups was determined using a paired *t*-test or a one-way repeated measures ANOVA and Tukey’s post-hoc test. *p* values of <0.05 (*), <0.01 (**) and <0.001 (***) were considered statistically significant.

## 3. Results

### 3.1. Recovery Effect on Neomycin-Induced Otic Hair Cells Damage in Zebrafish

The effect of AVM and DKB122 on otic neuromast hair cell recovery was evaluated after ototoxic drug (i.e., neomycin) exposure using a zebrafish model. The hair cells within the otic (O1) neuromast were severely damaged under neomycin exposure (Figure 1). Neomycin significantly decreased (*p* < 0.001) the number of otic hair cells. However, 10 µg/mL of AVM or DKB122 exhibited significant hair cell recovery (*p* < 0.01 and *p* < 0.001) compared to the control. Moreover, the DKB122 group demonstrated significant (*p* < 0.05) recovery in hair cells compared to AVM. Thus, DKB122 exhibited an increased efficacy on otic hair cell recovery after neomycin-induced ototoxicity.

### 3.2. EC_50_ Values of DKB122

We generated a dose-effect curve using zebrafish treated with DKB122 at six different DKB122 concentrations to evaluate the EC_50_. The EC_50_ values of DKB122 was calculated at 0.44 μg/mL (Figure 2).

### 3.3. Auditory Evaluation

We evaluated the effect of DKB122 on auditory sensory transmission using the ABR test. The baseline ABR thresholds before noise exposure were considered normal across the frequency range in each group. The threshold shifts of each group were elevated on day 1 after noise exposure. However, after 10 days of treatment, the 100 mg/kg DKB122 group significantly decreased the threshold shift in response to 8 kHz and 16 kHz stimuli (*p* < 0.05, *p* < 0.01 respectively) compared to the control; similarly, 300 mg/kg DKB122 significantly decreased the threshold shift in response to the click as well as the 8-kHz and 16-kHz stimuli (*p* < 0.05, *p* < 0.01 and *p* < 0.001 respectively) compared to the control. Furthermore, after 20 days of treatment, both DKB122 treatments showed the same tendency. Thus, DKB122 caused a significant improvement on auditory function in a dose-dependent manner and its efficacy is shown mainly at 8 and 16 kHz frequencies (Figure 3). These results indicate that DKB122 improves auditory sensory transmission function in a mice model of noise induced-hearing loss.

### 3.4. Otoprotective Effects of Cochlear Hair Cells

We focused in the mid-apex section of the cochlea. According to our ABR test, we evaluated the DKB122 effect at 8 and 16 kHz, which corresponds to that region of the cochlea [13,14,15]. That is the reason for us evaluating morphological changes specifically in that region. As shown in Figure 4A, there was noise-induced severe loss of OHCs and abnormalities as compared to the control animals (NIHL). The administration of DKB122 after noise-induced hearing loss largely preserved hair cells from noise-induced damage. As shown in Figure 4B, quantification of the OHC number showed significantly more OHCs in the DKB122-treated mice than in the no-treatment group (*p* = 0.0109). Our results indicate that DKB122 has therapeutic potential to treat noise-induced damage in hair cells.

### 3.5. Differential Gene Expression Induced by Neomycin in HEI-OC1 Cells

To investigate the protective mechanism of DKB122 on neomycin-induced inner ear cell damage, we sought to identify genes affected by neomycin using transcriptome analysis. To determine the optimal concentration of neomycin-dependent cell viability and cell damage, HEI-OC1 cells were treated with various concentrations of neomycin for 24 h. As the concentration of neomycin increased, the cell viability of HEI-OC1 cells decreased. The neomycin concentration that resulted in 20–30% cell death was determined to be the optimal concentration for this study (Figure 5A). RNA sequencing was performed to monitor differential gene expression in the genome in neomycin-treated HEI-OC1 cells. Of the 5574 genes expressed in HEI-OC1 cells, 143 genes were differentially expressed (FDR <0.05, |FC| > 2.0) by neomycin treatment and the results are presented as a scatter plot and heat map (Figure 5B,C). Of these 143 genes, 78 were up-regulated and 65 genes were down-regulated by neomycin treatment (Figure 5D).

Next, KEGG analysis was performed to identify the functional pathways of these differentially expressed genes. The genes that were up-regulated by neomycin treatment were enriched in the FoxO signaling pathway and transforming growth factor-beta (TGF-β) signaling pathway (Figure 5E). The FoxO signaling pathway is involved in a wide range of cell functions, including cellular differentiation, apoptosis, cell proliferation, and oxidative stress [16]. TGF-β signaling regulates cytokine secretion [17]. The genes that were down-regulated by neomycin treatment were enriched in glycine, serine, and threonine metabolism pathways and the biosynthesis of amino acids (Figure 5F). The effects of aberrant biosynthesis and metabolism of amino acid on cell death have been extensively studied. PSAT1 (phosphoserine aminotransferase 1), an enzyme catalyzing serine biosynthesis, promotes cell cycle progression and proliferation by inhibition of cyclin D1 degradation [18,19]. SHMT2 (Serine hydroxymethyltransferase) catalyzes the folate-dependent serine/glycine interconversion. Depletion of human SHMT2 induces apoptosis in U2OS cells [20]. Human CTH (cystathionase) plays an anti-apoptotic role in MCF-7 cells by promoting endoplasmic reticulum and mitochondrial homeostasis [21]. PSPH (phosphoserine phosphatase) expressed in the proliferative layer of the epidermis and hair follicles promotes cell proliferation [22]. These findings are consistent with the results of HEI-OC1 cell death by neomycin and suggest that defects in amino acid metabolism may be a major mechanism for HEI-OC1 cell death by neomycin.

### 3.6. DKB122 Rescues Altered Gene Expression and Reduced Viability by Neomycin in HEI-OC1 Cells

Next, we examined the effect of DKB122 on the expression of the genes affected by neomycin as identified by RNA-seq. When HEI-OC1 cells were treated with DKB122, the expression of genes increased by neomycin treatment that were related to the FoxO signaling pathway and the TGF-β signaling pathway (*Foxo1*, *Tgfb2*, and *Gadd45g*) were significantly inhibited (Figure 6A). These results suggest that DKB122 attenuates the signaling pathway related to apoptosis and cytokine secretion, which are increased by neomycin. In contrast, the expression of genes in pathways related to the biosynthesis of amino acids and the glycine, serine, and threonine metabolism pathways (*Cth*, *Psph*, *Psat*, and *Mthfd2*), which were all reduced by neomycin, were significantly rescued by DKB122 treatment (Figure 6B). As described above, these enzymes involved in amino acid biosynthesis and metabolism play an important role in cell proliferation by regulating cell cycle regulation and mitochondrial homeostasis [18,19]. Given that these gene expression changes have been restored by treatment with DKB122, our results show that DKB122 protects HEI-OC1 cells from the altered amino acid biosynthesis and metabolism pathway by neomycin, resulting in the inhibition of apoptotic cell death. Consistently, DKB122 treatment significantly inhibited neomycin-induced cell death of HEI-OC1 cells (Figure 6C).

## 4. Discussion

In this study, we focused on avocado oil as a source of fatty acids and several bioactive compounds with pharmacological potential [7]. In SNHL pathology, prevention is still the main goal to attenuate sensory hair cell loss [2] and omega-3 fatty acids have demonstrated efficacy for preventing age-related hearing loss previously [23,24]. Avocado oil is a functional food that contains linolenic acid, which is included in the omega-3 fatty acids group [25]. Thus, we investigated the efficacy of avocado oil on sensorineural hearing loss in vitro and in vivo and elucidated its mode of action. Our preliminary data showed the potential of AVM in otic hair cell recovery after damage by neomycin exposure in zebrafish. These results led us to develop an enhanced functional avocado oil (DKB122) through extraction with isopropyl alcohol, which is a well-known process to extract fatty acids from raw oils [26,27]. Then, we confirmed the effect of DKB122 on otic hair cell recovery and also compared its efficacy to AVM to establish enhanced activity. The EC_50_ value of DKB122 was 0.44 μg/mL in neomycin-induced zebrafish. Moreover, we evaluated the effects of DKB122 on auditory function in a mouse model of NIHL by ABR test and morphological evaluation of cochlear hair cells by rhodamine phalloidin staining. We next evaluated its protective effect in vitro using the HEI-OC1 cell line. To further investigate the mode of action of DKB122, we performed RNA sequencing and KEGG pathway enrichment analysis of DEGs. Finally, we analyzed the effects of DKB122 treatment on neomycin-induced changes in gene expression. Thus, we demonstrated that DKB122 can treat hearing loss, and its efficacy can be attributed to a reduction in altered gene expression related to oxidative stress, cytokine production and protein synthesis pathway.

In recent years, the use of in vitro and in vivo models have played a key role for studying SNHL. In this regard, auditory cell lines such as HEI-OC1 cells and the lateral line of zebrafish have been important to elucidate underlying mechanisms of the pathology, to find possible treatment targets and to test new molecules with potential activity for treating SNHL [2,28,29]. Thus, we evaluated avocado oil extracts on otic hair cell recovery after neomycin-induced ototoxicity in zebrafish. Ototoxic insult such as neomycin damages sensory structures in a similar mechanism to that of NIHL [30]. Our results revealed that DKB122 has a greater efficacy on hair cell recovery compared to avocado oil itself. Then, we focused on DKB122 to further confirm its efficacy on different forms of SNHL.

We further assessed the protective properties of DKB122 on sensory hair cells. The HEI-OC1 mouse cell line is important for studing prevention of cell death because mammalian hair cells do not regenerate, as its homologous among fish, birds, and amphibians. Moreover, neomycin-induced ototoxicity is well established using this cell line, and our data are consistent with previous reports of neomycin damage [30]. As we hypothesized, DKB122 showed protective efficacy on sensory hair cells damaged by neomycin. These results are supported by different studies that showed efficacy of omega-3 fatty acids to attenuate hearing loss [24,31].

One of the most prevalent forms of SNHL is caused by noise exposure [32]. Thus, we evaluated DKB122 activity on auditory function in a mouse model of NIHL. Using ABR, we showed that DKB122 has efficacy especially at 16 kHz in a dose-dependent manner. DKB122 significantly recovered the auditory function after 20 days of treatment and also showed a protective effect on mice sensory hair cells as shown using rhodamine phalloidin staining. This study thus shows compelling evidence both in vitro and in vivo that DKB122 is a potential treatment for SNHL.

Additionally, RNA-seq was performed to monitor the differential expression of genes after neomycin treatment and to identify the mode of action of DKB122 in neomycin-induced ototoxicity. The up-regulated genes by neomycin were enriched in the FoxO and TGF-β signaling pathways. The FoxO signaling pathway is involved in a wide range of cell functions such as oxidative stress and apoptosis [16]. Thus, the attenuation of this pathway suggests a potential mechanism of DKB122 for SNHL therapy [33]. Similarly, activation of TGF-β signaling, which regulates cytokine secretion [17], is also important in the pathogenesis of activation and progression of SNHL. As we expected, treatment with DKB122 attenuated the expression of the genes related to the oxidative stress and cytokine secretion signaling pathways (Foxo1, Tgfb2, Gadd45g). It has been reported that the suppression of Foxo1 can decrease oxidative stress, resulting in an improvement in cell survival [34]. Moreover, a number of attempts have been made to inactivate the TGF-β pathway as part of blocking apoptosis in a variety of pathological conditions [35] and Gadd45g has been reported to be involved in the apoptotic cascade [36].

In contrast, the expression of genes related to the biosynthesis of amino acids (Cth, Psph, Psat1, Mthfd2) was rescued by DKB122 treatment. The down-regulation of the genes involved in the biosynthesis of amino acids with the metabolic pathway of glycine, serine, and threonine by neomycin treatment implies the disturbance of intracellular catabolism and the assimilation process. A series of reports have shown that aminoglycoside drugs such as neomycin bind to the 12S subunit of mitochondrial rRNA and inhibit RNA translation [37,38,39]. However, this is limited to cases involving a specific mutation to mitochondrial 12S rRNA. Most recently, the inhibition of protein synthesis has been suggested as another mechanism of ototoxicity caused by aminoglycosides [40]. Consistently, our results also indicate that the synthesis and metabolism of amino acids may be affected by neomycin even in normal cells. Thus, it appears that DKB122 decreases cell death mediated by different pathways.

Taken together, our results demonstrate that DKB122 protects hair cells from neomycin by reducing the altered gene expression in the oxidative stress, cytokine production and protein synthesis pathways, which are significantly altered by neomycin. Future studies should focus specifically on the importance of fatty acids from avocado oil in different forms of SNHL (ototoxicity, NIHL).

## Figures and Tables

**Figure 1 nutrients-11-00113-f001:**
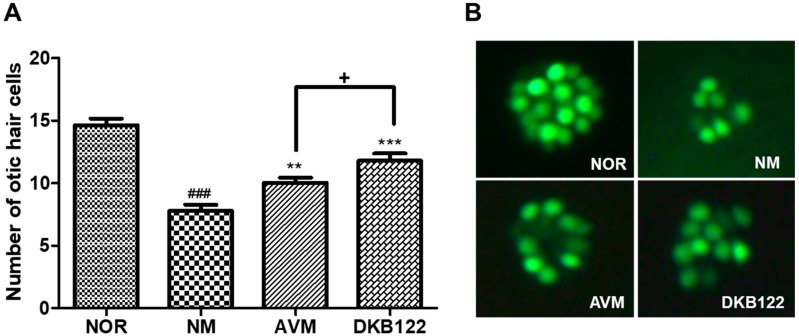
Otic hair cell recovery after neomycin-induced hair cell damage. (**A**) The number of otic hair cells in the untreated group (NM) and the treated groups: 10 µg/mL crude avocado oil (AVM), 10 µg/mL avocado oil extract (DKB122). (**B**) Fluorescence images of the zebrafish otic hair cells of the normal (NOR), control (NM), and treated groups. Hair cells stained with 0.1% YO-PRO-1. Data are presented as means ± SEM. + *p* < 0.05, ** *p* < 0.01, *** *p* < 0.001 (control versus treated groups). ### *p* < 0.001 (normal group versus control group).

**Figure 2 nutrients-11-00113-f002:**
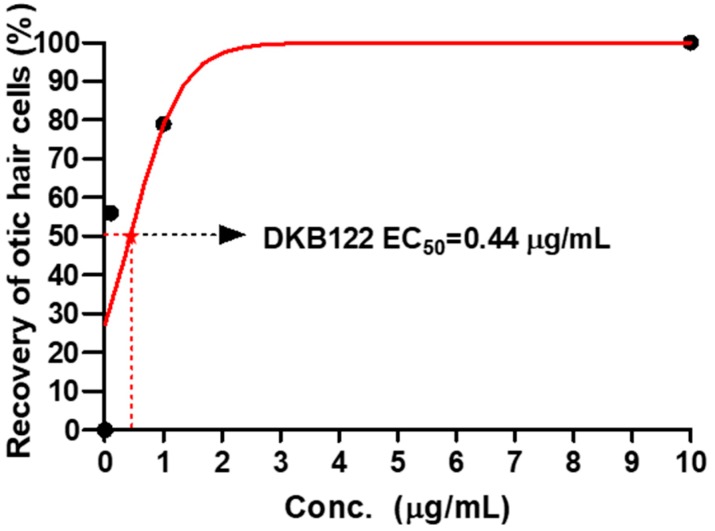
Dose-effect curves of DKB122. The EC_50_ of DKB122 was 0.44 μg/mL.

**Figure 3 nutrients-11-00113-f003:**
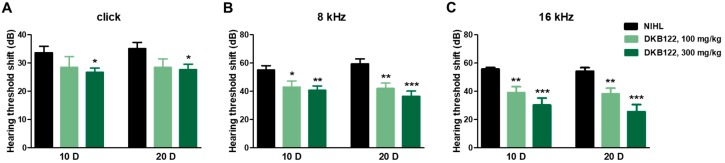
Effects of DKB122 on auditory function. Hearing thresholds shifts of Auditory Brainstem Response (ABR) with clicks (**A**), 8-kHz tone burst (**B**) and 16-kHz tone bust (**C**) in Noise-Induced Hearing Loss (NIHL) mice at 10 days (10 D) and 20 days (20 D) after noise-induced damage. Data are presented as means ± SEMs. * *p* < 0.05, ** *p* < 0.01, *** *p* < 0.001 (significant differences between the control and treated groups).

**Figure 4 nutrients-11-00113-f004:**
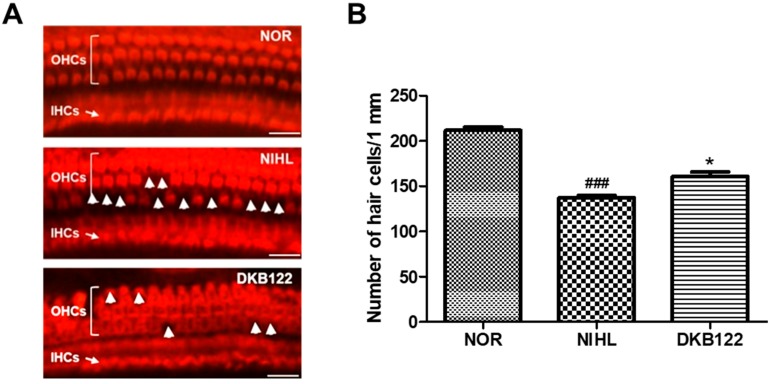
DKB122 alleviated noise-induced damage in cochlear hair cells. (**A**) Rhodamine phalloidin staining of the mid-apex section in the cochlea. Scale bar = 50 µm. (**B**) Number of OHCs in 1 mm (*n* = 6). Data are presented as means ± SEM. * *p* < 0.05 (NIHL versus treated groups), ### *p* < 0.001 (normal group versus NIHL group).

**Figure 5 nutrients-11-00113-f005:**
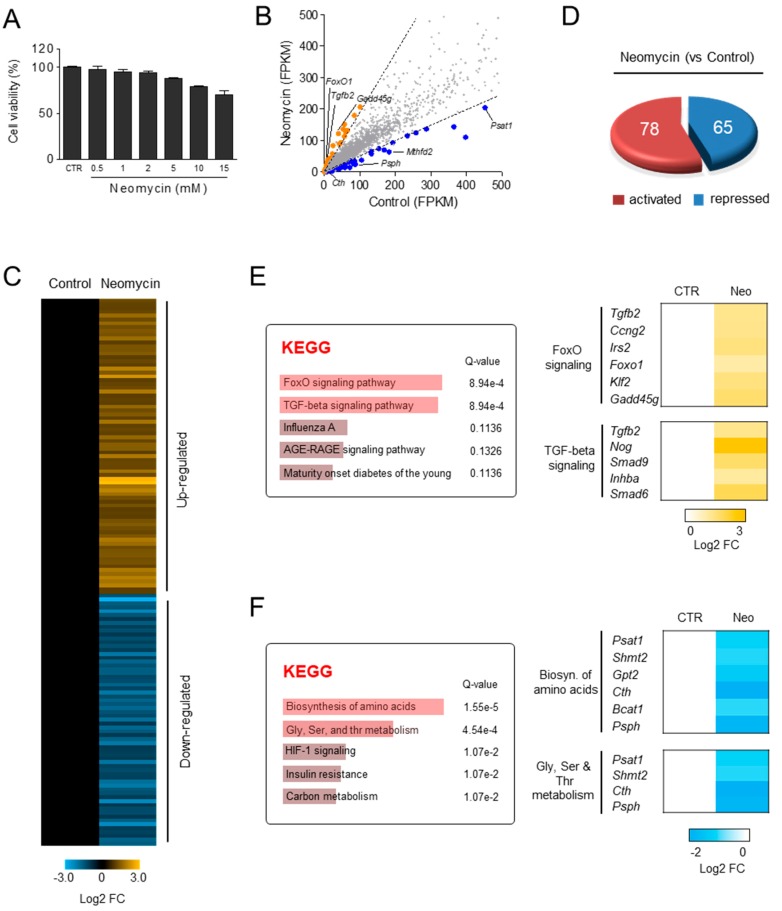
Differential gene expression induced by neomycin in House Ear Institute-Organ of Corti 1 (HEI-OC1) cells. (**A**) Viability of HEI-OC1 cells treated with neomycin (0.5–15 mM) for 24 h. (**B**) Scatter plot of control vs. neomycin (10 mM) group. Of the total genes, 143 were significantly altered by neomycin (FDR<0.05, |FC| > 2.0). (**C**) Heat map based on RNA-seq analysis of gene expression in HEI-OC1 cells. (**D**) Venn diagram showing the results of RNA-seq for the neomycin-regulated gene set. Genes were divided into neomycin-induced and neomycin-repressed gene groups. (**E**) Genes up-regulated by neomycin were categorized according to gene function by Kyoto Encyclopedia of Genes and Genomes (KEGG) analysis (Q < 0.05, left panel). Heat map generated using RNA-seq of gene sets in FoxO and TGF-β signaling (right panel). (**F**) Genes down-regulated by neomycin were analyzed by KEGG (Q < 0.05, left panel). Heat map of genes in amino acid biosynthesis and glycine, serine, and threonine metabolism (right panel).

**Figure 6 nutrients-11-00113-f006:**
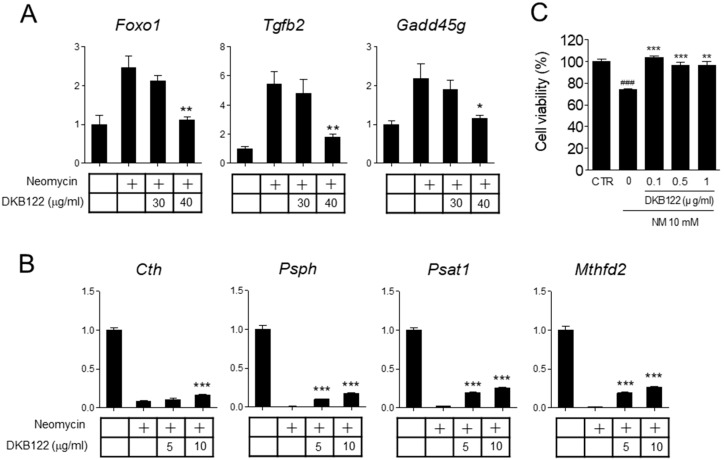
DKB122 rescued altered gene expression and improved viability in neomycin-treated House Ear Institute-Organ of Corti 1 (HEI-OC1) cells. (**A**,**B**) Effect of DKB122 on the expression of up-regulated (**A**) or down-regulated (**B**) genes was determined by RT-qPCR. HEI-OC1 cells were treated with neomycin (10 mM) for 24 h before harvesting. DKB122 (5–40 μg/mL) was added 1 h prior to neomycin treatment. Total RNA was analyzed by RT-qPCR. Levels of all mRNAs were normalized to that of 18S rRNA (*n* = 3). * *p* < 0.05; ** *p* < 0.01; *** *p* < 0.001 (Noise-Induced Hearing Loss (NIHL)) versus treated groups), ### *p* < 0.001 (normal group versus NIHL group). + Incubated with neomycin (10 mM) for 24 h. (**C**) Effect of DKB122 on the viability of neomycin-treated HEI-OC1 cells. Cells pretreated with DKB122 for 1 h were incubated with neomycin (10 mM) for 24 h.

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
