# Peer review of "Avocado Oil Extract Modulates Auditory Hair Cell Function through the Regulation of Amino Acid Biosynthesis Genes"

_nutrients, 2019, doi:10.3390/nu11010113_

Round 1
Reviewer 1 Report
The study by Nam et al. investigates protective effects of enhanced avocado oil extract against sensorineural hearing loss. The authors use in vitro and in vivo models to examine the protective effects of enhanced avocado oil extract against sound and neomycin-induced hair cell damage. The authors show that enhanced avocado oil extract treatment enhances otic hair cell recovery in zebrafish post neomycin-induced damage. The authors also show that enhanced avocado oil extract treatment improves auditory function and alleviates cochlear hair cells damage in mice post noise induced hair cell insult. Lastly, transcriptome data shows that enhanced avocado oil extract treatment reverses changes in expression levels of key differentially expressed genes between neomycin treated vs neomycin + enhanced avocado oil extract experimental groups.
Comments:
Line 30: add ‘the efficacy of avocado oil’
Line 37: add ‘House Ear Institute-Organ of Corti 1 (HEI-OC1)’
Always introduce abbreviations with full-form at first mention.
Line 56: add ‘efficacy of avocado oil’
Methods:
The authors have provided too little information about the ‘enhanced functional avocado oil extract or DKB122’. Add more details- like How was the extract prepared? What method was used to prepare the extract? and/or provide a valid reference which details the method.
What is the difference between crude avocado oil and enhanced functional avocado oil extract? Include the explanation in manuscript.
What was the rational for using specific concentrations (10 μg/mL for zebrafish experiments; 100mg/kg and 300mg/kg for NIHL mice experiments) of avocado oil extract for various in vitro or in vivo experiments?
Line 97-99: ‘To evaluate the effect of crude avocado oil (AVM), avocado oil extract (DKB122) on otic hair cell recovery, the neomycin solution was first rinsed with 0.03% sea salt solution and then the zebrafish were exposed to 10 μg/mL of AVM and DKB122 for 8 hours at 28°C.’ This is a very long and complex sentence which might confuse the readers. Re-phrase or break it down into two sentences.
Figure 2 lacks label A,B, and C. Change the colors of either 100mg/kg or 300mg/kg figure bars.
Line 246-267: ‘..suggest that defects in protein metabolism may be a major mechanism for HEI-OC1 cell death by neomycin.’ Protein metabolism is a very broad term and it poorly concludes the differential gene expression output. The authors do not offer any specific reasoning for a link between down regulation of glycine, serine, and threonine metabolism pathways and the biosynthesis of amino acids and neomycin induced cell death.
Line 269: ‘These results suggested that DKB122 protects HEI-OC1 cells from neomycin by reducing the altered gene expression in the oxidative stress, cytokine production, and protein synthesis pathway, which are all significantly changed by neomycin.’ Again this is a very broad and vague statement. Highlights poor synthesis of the data. Authors need to offer some rational for the observations made. Or present some logic by referencing literature which may help to explain why reduced gene expression of oxidative stress, cytokine production etc. correlates with a protective effect on DKB122.
The authors mention several times throughout the manuscript that avocado oil is a prominent source of fatty acids and several other bioactive compounds (potassium, vitamin B3 etc.), but never talk about their direct impact/role in protective effects against sound or neomycin induced hair cell damage.
Line 288: add ‘..efficacy of avocado oil on..’
Line 299: ‘Protecting sensory hair cells has become an important target in SNHL because it is a result of hair cell loss’. In this section of the manuscript, this sentence gives me no new information.
The title of the manuscript mentions ‘FoxO’, but there is no data showing the mechanism of regulation via FoxO in this manuscript. I feel the title is inappropriate representation of what the manuscript entails in its current form.
Author Response
Thank you for your comments.

Reviewer 2 Report
Methods:
Zebrafish—did the authors perform dose-response curves for DKB 122?
Did they use both crude avocado oil plus DKB 122? It appear that they did, based on the data shown for zebrafish in the results section. Please clarify.
In the mouse experiments, did they only test two doses of oral DKB 122?
Did they only test administration of DKB122 only at the one time point, one day after noise exposure?
Line 110-113. Were the animals treated for up to 20 days with DKB 122? Please clarify.
Why did they choose clicks instead of octave or broad band noise exposure? Recent papers on noise induced hearing loss in mice have employed either octave band or broad band noise exposures (see Amani pour et al, Noise-Induced Hearing Loss in Mice: Effects of High and Low Levels of Noise Trauma in CBA Mice. Cone Proc IEEE Eng. Med Boil Soc. 2018 Jul; 2018:1210-1213; Vlajkovic SM et al., Adenosine receptors regulate susceptibility to noise-induced neural injury in the mouse cochlea and hearing loss. Hear Res. 2017 Mar; 345:43-51; and Chen J et al, SHA Inhibitors of Histone Deacetylases Attenuate Noise-Induced
Hearing Loss JARO 17: 289–302 (2016).
Line 130, besides the technique used to measure click evoked thresholds the authors also tested thresholds for 8 and 16 kHz tone bursts. They should state that in addition to the threshold technique for click stimuli in their methods section.
Lines 139-140. The authors state that they only tested a 1 mm strip of the mouse cochlea to evaluate hair cells. Why did they only use a portion of the cochlea instead of the entire length of the cochlea? What part of the cochlea did they look at?
Line 157. What concentration of DKB 122 was applied for the mRNA analyses? On line 149, the authors list a range of concentrations.
Author Response
Thank you for your comments.

Round 2
Reviewer 1 Report
The revised manuscript successfully addresses the concerns from the previous review.
Reviewer 2 Report
The authors have responded well to concerns raised in the previous review.